# TESTA: Temporal-Spatial Token Aggregation
# for Long-form Video-Language Understanding

**Shuhuai Ren**[†], **Sishuo Chen**[§], **Shicheng Li**[†], **Xu Sun**[†], **Lu Hou**[‡]

[†]National Key Laboratory for Multimedia Information Processing,
School of Computer Science, Peking University

[§]Center for Data Science, Peking University [‡]Huawei Noah's Ark Lab

`shuhuai_ren@stu.pku.edu.cn` `{lisc99, chensishuo, xusun}@pku.edu.cn`
`houlu3@huawei.com`

## Abstract

Large-scale video-language pre-training has made remarkable strides in advancing video-language understanding tasks. However, the heavy computational burden of video encoding remains a formidable efficiency bottleneck, particularly for long-form videos. These videos contain massive visual tokens due to their inherent 3D properties and spatiotemporal redundancy, making it challenging to capture complex temporal and spatial relationships. To tackle this issue, we propose an efficient method called **TE**mporal-**S**patial **T**oken **A**ggregation (TESTA). TESTA condenses video semantics by adaptively aggregating similar frames, as well as similar patches within each frame. TESTA can reduce the number of visual tokens by 75% and thus accelerate video encoding. Building upon TESTA, we introduce a pre-trained video-language model equipped with a divided space-time token aggregation module in each video encoder block. We evaluate our model on five datasets for paragraph-to-video retrieval and long-form VideoQA tasks. Experimental results show that TESTA improves computing efficiency by 1.7 times, and achieves significant performance gains from its scalability in processing longer input frames, e.g., +13.7 R@1 on QuerYD and +6.5 R@1 on Condensed Movie.[1]

## 1 Introduction

Video-language modeling aims to learn semantic alignment between video and language in a joint representation space (Xu et al., 2021; Lei et al., 2021) to facilitate downstream tasks including text-video retrieval, video question answering (VideoQA), and video captioning. Unlike text, which can be represented concisely as a sequence of words with dense semantics, video input consists of much longer sequences due to its 3D properties and the redundancy in space-time information (He

et al., 2021; Tong et al., 2022). In fact, the number of visual tokens processed by Transformer-based models (Fu et al., 2021; Cheng et al., 2022; Ye et al., 2022; Li et al., 2021a; Wang et al., 2022b) can be over $150\times$ more than text tokens.[2] This poses an efficiency bottleneck for video-language understanding, especially for long-form videos lasting more than 30 seconds (Wu and Krähenbühl, 2021; Sun et al., 2022).

To encode long videos within limited computing budgets, previous approaches can be broadly categorized into two types: **(1) Sparse Sampling** (Lei et al., 2021; Sun et al., 2022; Lei et al., 2022). This method reduces the number of visual tokens by sampling very few frames from the raw video.[3] However, sparse sampling sacrifices rich temporal dynamics and storyline information, which limits model performance. **(2) Offline Encoding** (Luo et al., 2021; Bain et al., 2022). It allows processing more frames within the same computation budgets by constraining the interaction between visual tokens. It first uses an off-the-shelf image encoder (Dosovitskiy et al., 2020; Radford et al., 2021) to encode each frame independently, then uses a temporal module to aggregate all the frame features. However, the frame features encoded offline may not be well adapted to downstream tasks in various domains. Additionally, the post-aggregation mechanism also prohibits the full fusion of frame features (Cheng et al., 2022). Considering that both **sufficient input frames** and **full temporal-spatial modeling in an end-to-end manner** are pivotal for optimal performance, a natural question arises: *Are there better approaches to achieve efficient video coding without compromising on either of these aspects?*

---

[2]For example, in the QuerYD dataset, a long-form video with 96 sampled frames at a resolution of $224 \times 224$ pixels generates around **19K** visual tokens after patchification, while the corresponding caption contains only **128** text tokens.

[3]For instance, sample **4** frames from more than **5.4K** frames for ActivityNet Captions dataset (Krishna et al., 2017).

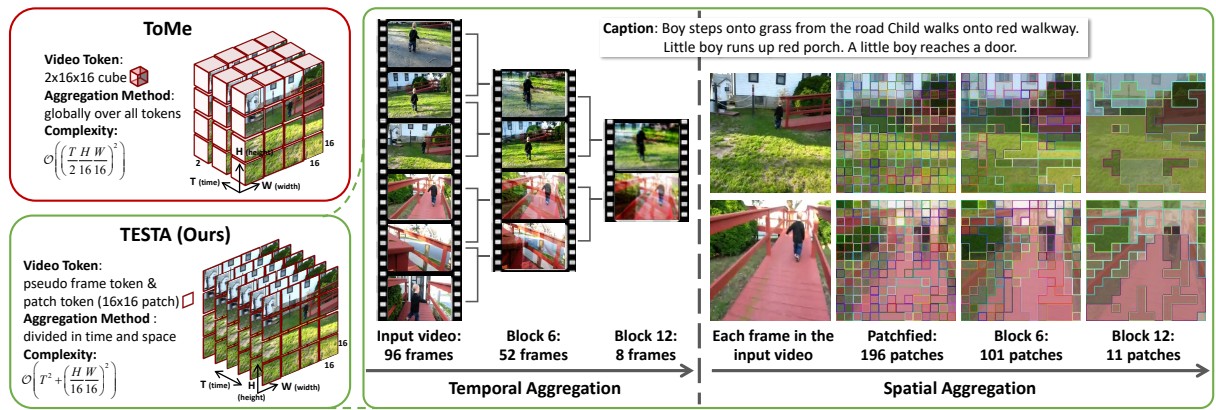

Figure 1: Two blocks on the left compare ToMe (Bolya et al., 2022) and our TESTA on three aspects: video token definition, aggregation method, and computation complexity. The block on the right illustrates TESTA's divided temporal aggregation (left) and spatial aggregation (right). Patches sharing the same inner and border colors are merged together. Our aggregation gradually reduces the number of frames and patches by averaging their features during the forward process of video encoding.

In this paper, we propose an efficient method named **TE**mporal-**S**patial **T**oken **A**ggregation (TESTA) inspired by Token Merging (ToMe) (Bolya et al., 2022). Specifically, TESTA samples input frames densely, but progressively aggregates similar visual tokens during video encoding to reduce the token number and computational overhead. As shown in Fig. 1, our aggregation operates separately in temporal and spatial dimensions, allowing for the merging of similar frames as well as similar patches within each frame. This reduces ToMe's complexity from $\mathcal{O}((\frac{T}{2}\frac{H}{16}\frac{W}{16})^2)$ to $\mathcal{O}(T^2 + (\frac{H}{16}\frac{W}{16})^2)$, making it more efficient for encoding longer videos. After aggregation, around 75% visual tokens can be reduced and thus the video encoding is accelerated. To achieve this, we use the bipartite matching algorithm. Specifically, we select a set of tokens and then find their most similar counterparts from the remaining set. Finally, we aggregate the features of these pairs through mean pooling. This aggregation-based mechanism has three advantages: **First**, it does not incorporate additional parameters and is amenable to parallelism, which significantly improves the training and inference efficiency; **Second**, our method (1) adaptively condenses video semantics rather than directly discarding input information, (2) retains full end-to-end spatiotemporal fusion, which both ensure the performance. **Third**, compared to convolution-based feature down-sampling methods (Liu et al., 2021; Li et al., 2021c), our aggregation trajectory can be easily tracked and

recovered. The aggregated tokens often correspond to higher-level semantics (e.g., objects, scenes, and events), making them more interpretable and even grounded in language.

Building upon TESTA, we design a pre-trained video-language model with a temporal and spatial token aggregation module in each video encoder block. We evaluate our model on paragraph-to-video retrieval and long-form VideoQA tasks. When using an equal number of input frames, our model improves computing efficiency by 1.7 times while maintaining comparable performance. When accessing more frames, our model exhibits strong scalability and achieves significant performance gains compared to previous state-of-the-art methods (e.g., +13.7 R@1 on QuerYD and +6.5 R@1 on Condensed Movie).

## 2 Related Work

**Video-Language Pre-trained Models.** Benefitting from large-scale video-text datasets (Bain et al., 2021; Xue et al., 2021) and advances in Transformer model design (Gorti et al., 2022; Ren et al., 2021; Fu et al., 2021; Zellers et al., 2021; Wang et al., 2022a), pre-trained Video-Language Models (VidLMs) (Chen et al., 2022; Sun et al., 2022; Cheng et al., 2022) have demonstrated impressive performance in video-language understanding tasks. VidLMs typically comprise a video encoder and a text encoder, which encode video-text pairs into a shared feature space to learn the semantic alignment between video and language. Additionally, a text decoder can be added after the video

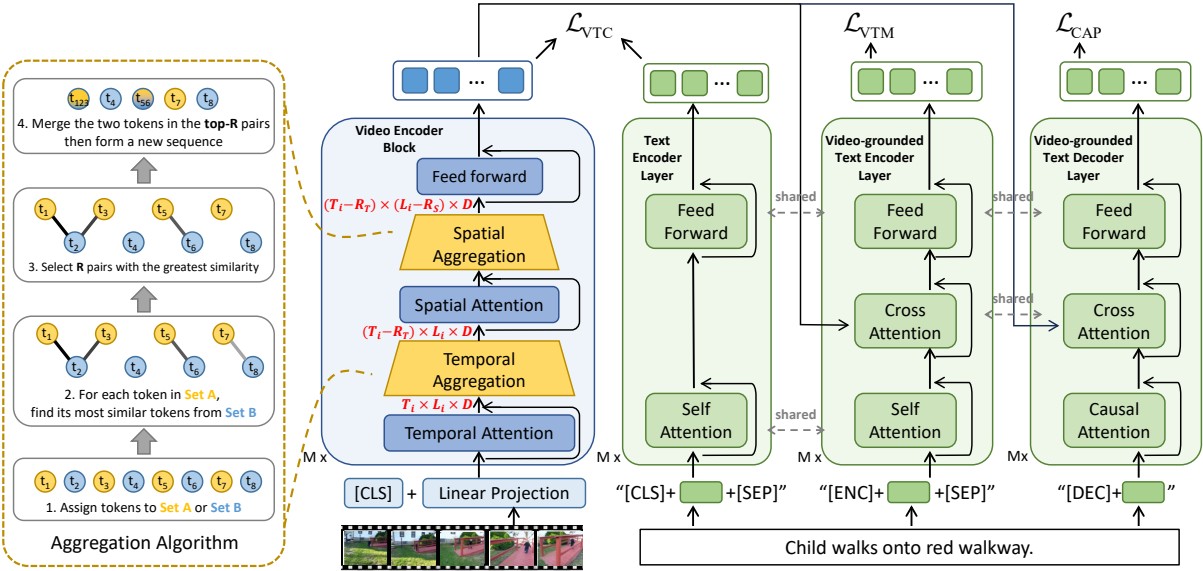

Figure 2: Architecture of our pre-trained model and token aggregation algorithm of TESTA. We record the size of the input and output features in red. The circles in the left panel denote either patch tokens or frame tokens.

encoder for tasks such as video captioning and VideoQA (Yan et al., 2022; Zhang et al., 2020).

**Efficient Video Transformer.** A Transformer-based video encoder typically pachifies each video into massive visual tokens, which will cause prohibitive computation costs for full self-attention with quadratic computational complexity. Therefore, research on efficient video Transformers has always been active. Representative work like TimeSFormer (Bertasius et al., 2021) and ViViT (Arnab et al., 2021) propose to factorize the spatial and temporal dimensions of the input, then separately apply spatial and temporal attention. Video Swin Transformer (Liu et al., 2021) keeps the joint temporal-spatial attention but restricts it within a local 3D window. Orthogonal to the advances of efficient Transformer architectures, our TESTA aggregates token features from the spatial and temporal dimensions, which reduces the size of input features for each Transformer block and can further boost the efficiency of video encoding.

**Feature Aggregation in Video Transformers.** Existing feature aggregation methods can be broadly categorized into two branches. Temporally, frame features can be encoded by a pre-trained image encoder and aggregated using self-attention, joint-attention, or mean pooling for post-temporal modeling purposes (Bain et al., 2022; Luo et al., 2021). Spatially, previous work explored merging similar patches in the image or aggregating tokens into additional proxy tokens (Bolya et al., 2022; Shi

et al., 2023; Cao et al., 2023; Xu et al., 2022; Ryoo et al., 2021; Marin et al., 2021). In contrast, we propose a unified mechanism to simultaneously aggregate frames and patches. Our method gradually aggregates features during video encoding, improving efficiency while ensuring sufficient interaction between features in both space and time.

## 3 Method

In this section, we first introduce our video-language pre-trained model and its architecture in § 3.1. To improve the efficiency of encoding long-form videos, we propose a novel temporal-spatial token aggregation mechanism (§ 3.2). Finally, we present the pre-training objectives in § 3.3.

### 3.1 Model Architecture

Inspired by prevalent VidLMs (Li et al., 2022, 2021b), our model consists of three encoders and one decoder for video-language representation learning. Figure 2 shows the model architecture.

**Text Encoder.** The text encoder is a uni-modal encoder similar to BERT (Devlin et al., 2019). A [CLS] token is prepended at the beginning of the input text to represent its global feature.

**Video-grounded Text Encoder.** This is a cross-modal encoder. Compared to the uni-modal text encoder, we add a cross-modal module to each encoder layer to enable information flow from video to language. We insert an [ENC] token before the

input text to condense the cross-modal information from both video and language.

**Video-grounded Text Decoder.** This is a cross-modal decoder with causal self-attention for auto-regressive text generation.

**Video Encoder.** This is a uni-modal encoder. Given a raw video, the visual input $V \in \mathbb{R}^{T \times H \times W \times 3}$ is a sequence of $T$ RGB frames of size $H \times W$ sampled from the video. Each frame is split into $L$ non-overlapping patches[4] following ViT (Dosovitskiy et al., 2020). To represent the global video feature, an additional [CLS] token is also used. Our video encoder is similar to TimeS-Former (Bertasius et al., 2021) with the Divided Space-Time Attention. Specifically, each video encoder block captures the temporal relations across frames using Temporal Attention and fuses the spatial information of objects, scenes, etc., within each frame using Spatial Attention. In contrast to TimeS-Former, we improve the efficiency of video encoding by equipping each video encoder block with a Temporal Aggregation Module and a Spatial Aggregation Module, which we will introduce in § 3.2.

## 3.2 Temporal-Spatial Token Aggregation

Videos have heavy spatiotemporal redundancy (He et al., 2021; Tong et al., 2022). On one hand, some activities (e.g., conversations) can persist across multiple frames with little visual variations. On the other hand, some scenes like background often contain numerous indistinguishable patches in each frame. Aggregating these similar frames and patches can simplify video feature representation and accelerate video encoding.

Accordingly, we introduce a Temporal Aggregation Module (TAM) and a Spatial Aggregation Module (SAM), i.e., the yellow modules in Figure 2. After each aggregation, TAM reduces $R_T$ frames while SAM reduce $R_S$ patches, where $R_T$ and $R_S$ are hyper-parameters to control the trade-offs between performance and efficiency. TAM and SAM are incorporated into each block of the video encoder, aggregating tokens progressively to reduce their number. For the $i$-th Transformer block, let $\mathbf{V} \in \mathbb{R}^{T_i \times L_i \times D}$ represents the input video feature, where $T_i, L_i, D$ denote the number of frames, the number of patches per frame, and the dimension of the token feature, respectively. The output video feature after temporal and spatial token aggregation is $\mathbf{V}' \in \mathbb{R}^{(T_i - R_T) \times (L_i - R_S) \times D}$, resulting in a smaller size and reducing the computing burden for subsequent blocks. After the forward process with $M$ encoder blocks, the final number of visual tokens is reduced to $(T - MR_T) \times (L - MR_S)$.

### 3.2.1 Objects for Aggregation

Our video encoder based on TESTA involves two types of tokens for aggregation: *patch tokens* and *frame tokens*. Recall that each frame is divided into a sequence of patches, which are treated as patch tokens. To ensure a formally unified aggregation algorithm, we define frame tokens as pseudo tokens to represent each frame by averaging all the patch tokens within it. When merging two frame tokens, the corresponding $L$ patches $[\mathbf{p}_1^{(1)}, \dots, \mathbf{p}_L^{(1)}]$ in frame-1 and $L$ patches $[\mathbf{p}_1^{(2)}, \dots, \mathbf{p}_L^{(2)}]$ in frame-2 are merged, resulting in $L$ patches $[\mathbf{p}_1^{(1\&2)}, \dots, \mathbf{p}_L^{(1\&2)}]$. As our aggregation strategy is agnostic to the token type, we refer to both patch tokens and frame tokens as "tokens" throughout the rest of the paper, without loss of generality.

### 3.2.2 Aggregation Strategy

Recall that given a sequence of $N$ tokens, our target is to reduce $R$ tokens after each aggregation operation.[5] To achieve this, we can greedily merge two tokens with the highest similarity and then repeat $R$ times, or merge $N$ tokens into $N - R$ clusters using clustering algorithms such as k-means (Lloyd, 1982). However, these iteration-based methods are not suited for parallelism and can slow down encoding speed (Bolya et al., 2022). Therefore, we resort to the bipartite matching method. We first partition the $N$ tokens into two disjoint sets $\mathbb{A}$ and $\mathbb{B}$ with $R$ and $N - R$ tokens, respectively. The $R$ tokens in the set $\mathbb{A}$ are selected elaborately as the tokens to be reduced. For each token in the set $\mathbb{A}$, we find its most similar token from the set $\mathbb{B}$, then merge them by averaging their features. As a result, the remaining $N - R$ tokens in the set $\mathbb{B}$ form a new sequence as the output.

For similarity calculation, we utilize the attention keys (K) of tokens as features and measure their similarity using cosine similarity. The attention keys contain summarized information intended for use in QKV self-attention, yielding accurate similarity measures (Bolya et al., 2022).

---

[4]The size of each patch is $P \times P$, and the $L$ patches span the entire frame ($L = HW/P^2$).

[5]For temporal aggregation, $N = T$ and $R = R_T$, for spatial aggregation, $N = L$ and $R = R_S$.

In practice, we introduce two aggregation algorithms, i.e., *importance-based* aggregation and *geometry-based* aggregation.

**Importance-based Aggregation.** In this algorithm, we pick out the least important $R$ tokens into the set $\mathbb{A}$ for aggregation, so as to minimize the negative effects of token reduction. The importance of the token $x_i$ is measured by the following score function $S_i$, which is defined as the product of the attention it receives from the other tokens $\sum_{j=1, j \neq i}^{N} \mathbf{A}_{ji}$:

$$S_i = \sum_{j=1, j \neq i}^{N} \mathbf{A}_{ji} = \sum_{j=1, j \neq i}^{N} \text{softmax}(\frac{\mathbf{Q}\mathbf{K}^\top}{\sqrt{d}})_{ji}, \quad (1)$$

where $\mathbf{A}_{ji}$ is the attention score from token $x_j$ to $x_i$, $\mathbf{Q}$ and $\mathbf{K}$ represent Queries and Keys in self-attention, respectively.

**Geometry-based Aggregation.** In practice, we notice that adjacent tokens have a larger similarity and should be merged. However, these adjacent tokens also have similar importance scores and thus are prone to be grouped into the same set in importance-based strategy, which hinders their aggregation. To address this issue, we partition the $N$ tokens in an alternative way inspired by Bolya et al. (2022), thus assigning adjacent tokens to different sets $\mathbb{A}$ and $\mathbb{B}$. As shown in the left panel in Figure 2, for each token $x_i^{(\mathbb{A})}$ in the set $\mathbb{A}$, we find its most similar token $x_j^{(\mathbb{B})}$ from the set $\mathbb{B}$ to construct a pair $(x_i^{(\mathbb{A})}, x_j^{(\mathbb{B})})$ and record their similarity. After that, we select $R$ pairs with the greatest similarity and merge the two tokens in the top-$R$ pairs. Finally, we concatenate the tokens in the two sets back into one sequence as the output.

The above aggregation algorithms are parameter-free, and can be easily plugged into a Transformer-based video encoder. We conduct our aggregation during both training and testing. Although the token similarity calculation brings additional computing overhead, it is negligible compared to the efficiency gained by reducing token numbers.

### 3.2.3 Novelty over Token Merging

Our work is inspired by Token Merging (ToMe) (Bolya et al., 2022), which also proposes to reduce video tokens by merging similar ones. However, we differentiate ourselves from ToMe in two significant ways:

**Video Token Definition.** ToMe uses joint space-time tokens ($2 \times 16 \times 16$ cubes), while our TESTA defines frame tokens (representing entire frames) and patch tokens ($16 \times 16$ 2D patches) for decoupled aggregation. This tailored token design is more efficient for modeling long-form videos.

**Aggregation Method.** ToMe performs global aggregation over all tokens, resulting in a complexity of $\mathcal{O}((\frac{T}{2}\frac{H}{16}\frac{W}{16})^2)$. This becomes impractical for long-form video and causes out-of-memory issues beyond 16 frames. In contrast, TESTA uses divided aggregation in time and space, reducing complexity to $\mathcal{O}(T^2 + (\frac{H}{16}\frac{W}{16})^2)$. This allows efficient encoding of much longer videos (more than 128 frames under the same computation quota). The divided scheme also better captures spatial and temporal semantics, resulting in improved performance on long-form video understanding tasks (to be shown in § 4.7).

### 3.3 Pre-training Objectives

We use the following three classic pre-training objectives, i.e., video-text contrastive loss, video-text matching loss, and captioning loss. Please refer to Appendix A for more details.

## 4 Experiments

### 4.1 Implementation Details

To pre-train our TESTA model, we start by initializing it with the BLIP (12-layer ViT-B/16) checkpoint (Li et al., 2022), with the exception of the temporal attention, which is copied from the spatial attention weights. We use around 5M image-text and video-text pairs from two datasets for pre-training. See Appendix A for more details.

For downstream fine-tuning, we uniformly sample either 32 or 96 frames, each with a resolution of $224 \times 224$ pixels (196 patches per frame with a patch size of 16). To achieve approximately a $50\%$ reduction in computation cost, we employ different hyper-parameters for aggregation. Specifically, for 96-frame inputs, we set $R_T$ to 4 and $R_S$ to 8, while for 32-frame inputs, $R_T$ is 1 and $R_S$ is 12. We use geometry-based aggregation by default since it achieves better performance. Please refer to Appendix B for more fine-tuning details.

### 4.2 Downstream Task Setups

We finetune and evaluate TESTA on two downstream tasks of paragraph-to-video retrieval and

| Method | #PT Data | #Frame | GFLOPs ↓ | QuerYD | | | Condensed Movie | | |
|---|---|---|---|---|---|---|---|---|---|
| | | | | R@1 ↑ | R@5 ↑ | R@10 ↑ | R@1 ↑ | R@5 ↑ | R@10 ↑ |
| MoEE (Miech et al., 2018) | - | - | - | 11.6 | 30.2 | 43.2 | 1.9 | 7.8 | 13.4 |
| TeachText (Croitoru et al., 2021) | - | - | - | 14.4 | 37.7 | 50.9 | 12.1 | 27.4 | 37.5 |
| Frozen (Bain et al., 2021) | 5M | 32 | 1424 | 53.8 | 75.7 | 82.7 | - | - | - |
| LF-VILA (Sun et al., 2022) | 8M | 32 | **298** | 69.7 | 85.7 | 90.3 | 13.6 | 32.5 | 41.8 |
| VINDLU † (Cheng et al., 2022) | 25M | 32 | 745 | 67.8 | 86.3 | 81.8 | 18.4 | 36.4 | 44.3 |
| TESTA (Ours) | 5M | 32 | 420 | 77.0 | 91.3 | 94.6 | 21.5 | 42.4 | 50.7 |
| TESTA w/o agg. | 5M | 32 | 786 | 79.7 | 92.6 | 95.5 | 23.5 | 45.4 | 54.8 |
| TESTA (Ours) | 5M | 96 | 1381 | **83.4** | **93.8** | **95.3** | **24.9** | **46.5** | **55.1** |
| TESTA w/o agg. | 5M | 96 | 2383 | 84.2 | 93.8 | 95.1 | 25.5 | 46.8 | 56.0 |

Table 1: Paragraph-to-video retrieval performance (Recall@$k$) on QuerYD and Condensed Movie. **#PT Data** refers to the number of video-text pairs used for pre-training. † indicates the results of our re-implementation. **TESTA w/o agg.** denotes fine-tuning our pre-trained model without activating the token aggregation modules, resulting in no reduction in token number. This serves as an upper bound for TESTA's performance.

| Method | #PT Data | #Frame | GFLOPs ↓ | DiDeMo | | | ActivityNet Caption | | |
|---|---|---|---|---|---|---|---|---|---|
| | | | | R@1 ↑ | R@5 ↑ | R@10 ↑ | R@1 ↑ | R@5 ↑ | R@10 ↑ |
| TeachText (Croitoru et al., 2021) | - | - | - | 21.6 | 48.6 | 62.9 | 23.5 | 57.2 | - |
| ClipBERT (Lei et al., 2021) | 0.2M | 2 | 13 | 20.4 | 48.0 | 60.8 | 21.3 | 49.0 | 63.5 |
| Frozen (Bain et al., 2021) | 5M | 4 | 178 | 31.0 | 59.8 | 72.4 | - | - | - |
| LF-VILA (Sun et al., 2022) | 8M | 32 | 298 | 35.0 | 64.5 | 75.8 | 35.3 | 65.4 | - |
| ALPRO (Li et al., 2021a) | 5M | 8 | 197 | 35.9 | 67.5 | 78.8 | - | - | - |
| BridgeFormer (Ge et al., 2022) | 5M | 4 | 71 | 37.0 | 62.2 | 73.9 | - | - | - |
| Singularity (Lei et al., 2022) | 5M | 32 | 589 | 47.4 | 75.2 | 84.0 | 43.0 | 70.6 | 81.3 |
| HiTeA (Ye et al., 2022) | 5M | 12 | 98 | 51.8 | 79.1 | 85.3 | 45.1 | 73.5 | 84.2 |
| VINDLU (Cheng et al., 2022) | 5M | 4 | 93 | 54.6 | 81.3 | 89.0 | 51.1 | 79.2 | 88.4 |
| All-in-one (Wang et al., 2022a) | 138M | 3 | 62 | 32.7 | 61.4 | 73.5 | 22.4 | 53.7 | 67.7 |
| Clip4Clip (Luo et al., 2021) | 400M | 64 | 282 | 43.4 | 70.2 | 80.6 | 40.5 | 72.4 | - |
| X-CLIP (Ma et al., 2022) | 400M | 64 | 1086 | 47.8 | 79.3 | - | 46.2 | 75.5 | - |
| CLIP-ViP (Xue et al., 2022) | 100M | 12 | 212 | 50.5 | 78.4 | 87.1 | 53.4 | 81.4 | 90.0 |
| TESTA (Ours) | 5M | 32 | 420 | 57.7 | 83.3 | 89.4 | 51.7 | 79.1 | 87.6 |
| TESTA (Ours) | 5M | 96 | 1381 | **59.2** | **83.5** | **89.8** | **53.7** | **79.9** | **88.9** |

Table 2: Paragraph-to-video retrieval performance on DiDeMo and ActivityNet Caption. We gray out methods that use significantly more pre-training data for a fair comparison. The other notations are the same as those on Table 1.

| Method | #PT Data | Accuracy (%) |
|---|---|---|
| LF-VILA (Sun et al., 2022) | 8M | 39.9 |
| Singularity (Lei et al., 2022) | 5M | 41.8 |
| VIOLET (Fu et al., 2021) | 183M | 38.9 |
| JustAsk (Yang et al., 2020) | 69M | 38.9 |
| MERLOT (Zellers et al., 2021) | 180M | 41.4 |
| TESTA (Ours) | 5M | **45.0** |

Table 3: Accuracy (%) on ActivityNet-QA.

long-form VideoQA. For paragraph-to-video retrieval, we use four datasets: DiDeMo (Hendricks et al., 2017), QuerYD (Oncescu et al., 2020), ActivityNet Captions (Krishna et al., 2017), and Condensed Movie (Bain et al., 2020). For long-form VideoQA, we use ActivityNet-QA (Yu et al., 2019). The details of these datasets are shown in Appendix C.

### 4.3 Paragraph-to-Video Retrieval

Table 1 demonstrates the performance of TESTA on two challenging and under-explored paragraph-to-video retrieval datasets, QuerYD and Condensed Movie, which involve videos with lengthy durations (over 200 seconds on average). For 32-frame video inputs, TESTA achieves Recall@1 of 77.0 on QuerYD and 21.5 on Condensed Movie, surpassing previous SOTA methods by 7.3 and 3.1, respectively. In terms of computational complexity, TESTA exhibits a significantly lower GFLOPs of 420 compared to Frozen (Bain et al., 2021) and VINDLU (Cheng et al., 2022). While LF-VILA (Sun et al., 2022) operates with even fewer GFLOPs (298), it necessitates feature aggregation within a fixed local window, which can potentially undermine semantic integrity after concentration. In contrast, our model enables the adaptive merging of features with high similarity in the global scope, resulting in improved performance ($+7.6$

| Method | #PT Data | GFLOPs↓ | QuerYD | | | DiDeMo | | | ActivityNet Caption | | |
|---|---|---|---|---|---|---|---|---|---|---|---|
| | | | R@1↑ | R@5↑ | R@10↑ | R@1↑ | R@5↑ | R@10↑ | R@1↑ | R@5↑ | R@10↑ |
| Clip4Clip (Luo et al., 2021) | 400M | 282 | 50.0 | 74.5 | 83.3 | 43.6 | 71.3 | 79.0 | 25.0 | 51.6 | 65.7 |
| BLIP (Li et al., 2022) | 129M | 707 | 50.7 | 67.6 | 73.5 | 60.9 | 84.9 | 91.0 | 34.2 | 60.0 | 70.7 |
| TESTA (Ours) | 5M | 786 | **64.4** | **82.9** | **86.9** | **64.9** | **88.7** | **91.8** | **37.1** | **63.7** | **75.4** |

Table 4: Zero-shot evaluation (32 frames) on paragraph-to-video retrieval performance.

| TESTA | R@1↑ | R@5↑ | R@10↑ | Avg. ↑ | GFLOPs↓ | Memory (GB)↓ |
|---|---|---|---|---|---|---|
| No Aggregation | 84.2 | 93.8 | 95.1 | 91.0 | 2382.5 | 19.2 |
| *(1) Token Aggregation v.s. Token Pruning (w/o training for both)* | | | | | | |
| Token Pruning ($R_T = 4, R_S = 8$) | 71.0 | 86.1 | 90.6 | 82.6 | **1380.9** | **12.6** |
| Token Aggregation ($R_T = 4, R_S = 8$) | **79.2** | **91.8** | **95.3** | **88.8** | 1381.4 | **12.6** |
| *(2) Aggregation Strategy* | | | | | | |
| Importance-based Aggregation | 80.2 | 91.7 | 94.6 | 88.9 | **1380.9** | 13.7 |
| Geometry-based Aggregation | **83.4** | **93.8** | **95.3** | 90.8 | 1381.4 | **12.6** |
| *(3) Aggregation dimension* | | | | | | |
| Only temporal ($R_T = 7$) | 79.5 | 92.9 | **95.4** | 89.3 | **1303.9** | **11.5** |
| Only spatial ($R_S = 14$) | 81.4 | 93.3 | 95.1 | 89.9 | 1364.0 | 11.9 |
| Both temporal and spatial ($R_T = 4, R_S = 8$) | **83.4** | **93.8** | 95.3 | **90.8** | 1381.4 | 12.6 |

Table 5: Ablation study on (1) token reduction method, (2) aggregation strategy, and (3) aggregation dimension. The results are reported on QuerYD with 96 frames. Avg. represents average recall across R@1, R@5, and R@10.

R@1 on average compared to LF-VILA).

Given the importance of incorporating more input frames for long video understanding tasks, we finetune TESTA using 96-frame inputs and further promote R@1 to 83.4 on QuerYD and 24.9 on Condensed Movie. This exhibits strong scalability of our model (see Appendix D for a detailed analysis). Additionally, we report the results of TESTA without token aggregation, which serves as an upper bound for TESTA's performance. Although preserving full visual tokens yields higher recall, it requires 1.8 times more GLFOPs compared to TESTA. As the number of input frames increases from 32 to 96, the GFLOPs of TESTA w/o agg. exceed 2300, but the performance gain diminishes (only +0.8 R@1 on QuerYD). This indicates the superiority of our method in aggregating redundant tokens in long sequence inputs.

Table 2 demonstrates model performance on DiDeMo and ActivityNet Caption, which consist of shorter videos (∼100 seconds on average) and are considered less challenging. For 32-frame inputs, TESTA with 5M pre-training data achieves 57.7 R@1 on DiDeMo, which even surpasses the models pre-trained with over 100M data. By increasing the number of frames to 96, TESTA achieves R@1 of 59.2 on DiDeMo and 53.7 on ActivityNet, outperforming previous SOTA methods by 2.7 and 2.6, respectively.

### 4.4 Long-Form Video Question-Answering

Table 3 showcases the performance of TESTA on ActivityNet-QA (using 96-frame). The accuracy of TESTA is 45.0%, which is 3.2% higher than the previous SOTA, Singularity (Lei et al., 2022). This demonstrates that our method eliminates redundant information while integrating crucial visual cues to accurately answer the posed questions.

### 4.5 Zero-shot Generalizability

In Table 4, we show the zero-shot performance of pre-trained CLIP4clip, BLIP, and TESTA on three datasets (32 frames). Although our TESTA is initialized by the BLIP checkpoint, it consistently outperforms BLIP (as well as CLIP4clip) after our pre-training, achieving average improvements of +14.1, +2.9, and +3.8 on QuerYD, DiDeMo, and ActivityNet respectively. This indicates our substantial gains on long-form video datasets are not solely due to the strong BLIP checkpoint, but also owing to our temporal modeling and pre-training on video data.

### 4.6 Ablation Study

We perform an extensive ablation study and analysis on various crucial components in our aggregation algorithm to examine their impacts.

**Token Aggregation v.s. Token Pruning.** We first compare the performance and efficiency of

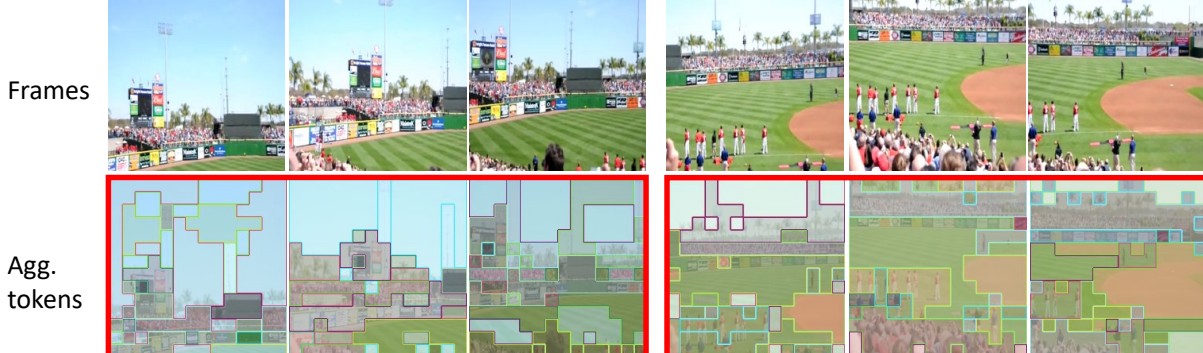

**Caption**: We see baseball park for the first time. The first time the baseball stadium is shown.

Figure 3: Visualization of our temporal and spatial aggregation. Frames that are enclosed within the same red rectangle, as well as patches that share the same inner and border color, are merged together.

token aggregation and token pruning (Rao et al., 2021). Regarding pruning, we calculate the importance score (Eq. (1)) for each token and prune the least important $R$ tokens following previous methods (Goyal et al., 2020). We finetune our pre-trained model on QuerYD without token aggregation, then apply token aggregation and pruning in an off-the-shelf manner for test evaluation. The results are presented in the first block of Table 5. In comparison to the vanilla model (no aggregation), both pruning and aggregation decrease computation costs, with only 58% GFLOPs and 66% GPU memory. However, the performance degradation of our token aggregation is much smaller than that of pruning (−2.2 v.s. −8.4 in terms of average recall), suggesting that aggregation better preserves the valuable visual semantics within videos.

**Ablation on the Aggregation Strategy.** To investigate the effectiveness of different aggregation strategies, we report the performance of TESTA using importance-based and geometry-based aggregation methods. The results in the middle block of Table 5 show that the simplest geometry-based aggregation method achieves the best Recall@1 of 83.4, outperforming the other method by 3.2. This confirms our hypothesis that adjacent tokens exhibit greater similarity and should be assigned to separate sets for aggregation.

**Ablation on the Aggregation Dimension.** We compare the performance of three aggregation methods: (1) temporal only, (2) spatial only, and (3) both temporal and spatial. To ensure a roughly equal computational overhead, we adjust $R_S$ and $R_T$ accordingly. The results in the bottom block of Table 5 show that performing token aggregation on

a single dimension leads to excessive dilution of information, while the information in other dimensions becomes overly redundant. This imbalance hurts the performance of the model. Therefore, our approach, with incorporates both temporal and spatial aggregation, achieves the best outcomes.

Additionally, Appendix E discusses the impact of the number of reduced tokens $R_T$ and $R_S$. Appendix F analyzes the properties of aggregated tokens by probing their similarity.

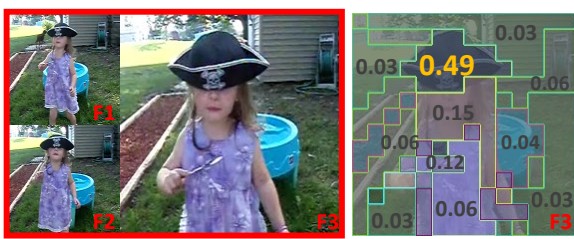

**S1:** A girl in **a pirate hat** walks towards the camera.

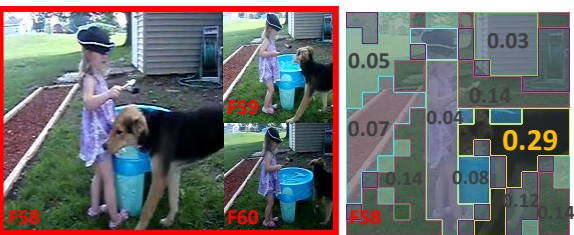

**S2:** **A dog** walks up to a little girl then walks away.

Figure 4: Text grounding visualization. F$i$ denotes the $i^{th}$ frame in the video and S$i$ denotes the $i^{th}$ sentence in the caption. We calculate the similarity between the phrase query (in orange) and each region formed by our aggregation, then record the value in the region. The phrase queries can be grounded to their corresponding aggregated regions, achieving the highest similarity.

| Method | GFLOPs ↓ | R@1 ↑ | R@5 ↑ | R@10 ↑ |
|---|---|---|---|---|
| ToMe | 252 | 59.9 | 82.2 | 88.6 |
| TESTA | **228** | **62.4** | **85.6** | **91.1** |
| ToMe w/o agg. | 450 | 66.1 | 86.4 | 90.4 |
| TESTA w/o agg. | **392** | **75.0** | **91.1** | **93.8** |

Table 6: Paragraph-to-video retrieval performance (Recall@$k$) on QuerYD (16 frames). **w/o agg.** denotes fine-tuning without token aggregation; the only distinction lies in the attention mechanism, where ToMe employs global attention, while TESTA utilizes separate spatial-temporal attention.

### 4.7 Comparison to Token Merging

We directly compare the performance of ToMe (Bolya et al., 2022) and TESTA by initializing both models from the BLIP pre-trained checkpoint and fine-tuning them on QuerYD. As we noted in § 3.2.3, due to the extremely high computational complexity of ToMe's global attention, increasing the number of input frames can lead to out-of-memory issues without token aggregation (w/o agg.). Therefore, we limit the number of input frames to 16. Besides, We set the hyperparameter $R$ (number of reduced tokens) to ensure matched GFLOPs. Specifically, for ToMe, $R = 197$, while for TESTA, $R_T = 1$ and $R_S = 2$. The results in Table 6 illustrate TESTA's efficiency and effectiveness for long-form video understanding, which can be attributed to our tailored design for divided spatial-temporal modeling. In comparison to ToMe, our approach achieves higher recall with fewer GFLOPs, regardless of whether token aggregation is applied.

### 4.8 Visualization

Figure 3 provides a visualization of temporal and spatial aggregation on the DiDeMo dataset. TESTA effectively aggregates tokens with highly-similar semantics, demonstrating its strong interpretability. **From a temporal perspective**, TESTA aggregates a sequence of frames captured during continuous lens movement (first 3 frames). It also condenses similar frames of athletes waiting for the game (last 3 frames). **From a spatial perspective**, TESTA merges the patches belonging to the same *scenes* (e.g., sky, baseball park) and the same *objects* (e.g., billboard, back of the audience's head). More examples can be found in Appendix G.

In Figure 4, we further show that TESTA enables grounding of language to the aggregated visual to-

kens (Ren et al., 2023b,a). Given the phrase query in the caption, it achieves the highest similarity of its oracle region formed by our aggregation, facilitating fine-grained alignment between phrases and regions.

## 5 Conclusion

In this paper, we present TESTA, an efficient method for long-form video-language understanding. By aggregating similar frames and patches, TESTA effectively condenses video semantics and accelerates video encoding. Experimental results on paragraph-to-video retrieval and VideoQA tasks demonstrate that TESTA outperforms previous SOTA methods by a considerable margin.

## Limitations

To facilitate future research, we analyze the limitations and possible solutions in our work. **(1)** Due to limited computing resources, we do not use long-form video pre-training datasets such as HD-VILA (Xue et al., 2021) or incorporate TESTA in pre-training. We believe long video pre-training with TESTA could greatly improve pre-training efficiency and obtain a video-language model with better performance. **(2)** For aggregation efficiency, we only use video-side features to merge visual tokens. We believe that leveraging text signals for aggregation could make the final encoded features more suitable for downstream tasks. **(3)** Our model training only uses coarse objectives such as VTC, VTM, and CAP (Eq. (2)-(4)) on video-text pairs. Considering TESTA can aggregate tokens into objects, scenes, events, etc., training with fine-grained alignment functions (Ren et al., 2021; Wang et al., 2022c) could help some tasks like action localization and video object detection (Zhukov et al., 2019; Real et al., 2017), on which we will perform more explorations in future work.

## Acknowledgements

We thank all the anonymous reviewers for their constructive comments, and Rundong Gao and Lei Li for their valuable suggestions in preparing the manuscript. This work is supported in part by a Huawei Research Grant and National Natural Science Foundation of China (No. 62176002). Xu Sun is the corresponding author of this paper.

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

# A Pre-training Details

## A.1 Pre-training Datasets.

We perform pre-training on two datasets: WebVid-2M (Bain et al., 2021) containing 2.5M video-text pairs and Conceptual Captions (CC3M) (Changpinyo et al., 2021) consisting of 3M image-text pairs. We include CC3M to improve spatial representations of videos as suggested by Li et al. (2021a). We duplicate images from CC3M for 8 times to make static videos. For WebVid-2M, we randomly sample 8 frames for each video instance. Because a small fraction of video and image URLs from the original datasets are no longer available, the total number of pre-training samples is around 5M. In the pre-training phase, we do not perform token aggregation since the number of frames in the pre-training video data is relatively small.

## A.2 Detailed Pre-training Objectives.

We use the following three classic pre-training objectives.

**Video-Text Contrastive Loss.** Given a batch of $B$ video-text pairs, the contrastive objective aims to pull together the paired videos and texts while pushing apart the others with dissimilar semantics in the feature space. Let $\mathbf{v}_i$ and $\mathbf{t}_i$ represent the [CLS] feature of the video and text, respectively.

| Dataset | Domain | #Video-Text Pairs | Avg. Len (sec) | Text Len | Duration (h) |
|---|---|---|---|---|---|
| WebVid-2.5M (Bain et al., 2021) | open | 2.5M | 18.0 | 12.0 | 13K |
| QuerYD (Oncescu et al., 2020) | open | 2K | 278.0 | 243.8 | 200 |
| Condensed Movie (Bain et al., 2020) | movie | 34K | 132.0 | 18.0 | 1.3K |
| DiDeMo (Hendricks et al., 2017) | Flickr | 10K | 28.0 | 29.2 | 87 |
| ActivityNet Captions (Krishna et al., 2017) | action | 20K | 180.0 | 48.3 | 849 |
| ActivityNet QA (Yu et al., 2019) | action | 5K | 117.0 | 8.9 | 976 |

Table 7: Statistics of video-language datasets.

The video-to-text contrastive loss $\mathcal{L}_{\mathrm{V2T}}$ is:

$$\mathcal{L}_{\mathrm{V2T}} = -\frac{1}{B}\sum_{i=1}^{B}\log\frac{\exp(\mathbf{v}_i^\top \mathbf{t}_i/\tau)}{\sum_j \exp(\mathbf{v}_i^\top \mathbf{t}_j/\tau)},$$

where $\tau$ is a learnable temperature parameter. Similarly, the text-to-video contrastive loss $\mathcal{L}_{\mathrm{T2V}}$ is:

$$\mathcal{L}_{\mathrm{T2V}} = -\frac{1}{B}\sum_{i=1}^{B}\log\frac{\exp(\mathbf{t}_i^\top \mathbf{v}_i/\tau)}{\sum_j \exp(\mathbf{t}_i^\top \mathbf{v}_j/\tau)}.$$

The video-text contrastive loss is defined as:

$$\mathcal{L}_{\mathrm{VTC}} = \frac{1}{2}(\mathcal{L}_{\mathrm{V2T}} + \mathcal{L}_{\mathrm{T2V}}). \qquad (2)$$

In the implementation $\mathcal{L}_{\mathrm{VTC}}$, the negative sample features are extracted from a queue of recent samples encoded by a momentum encoder (He et al., 2020). Moreover, a momentum distillation regularization loss (Li et al., 2021b) is added to $\mathcal{L}_{\mathrm{VTC}}$ for the sake of the potential positives in the negative pairs.

**Video-Text Matching Loss.** Video-text matching aims to predict whether a pair of video and text is matched or not. For the $i$-th video-text pair, we first obtain their joint video-text embedding of the [ENC] token from the video-grounded text encoder. We then use this embedding to generate a two-class probability $\mathbf{p}_i$, and calculate the video-text matching loss $\mathcal{L}_{\mathrm{VTM}}$ as:

$$\mathcal{L}_{\mathrm{VTM}} = \frac{1}{B}\sum_{i=1}^{B}\mathrm{CE}(\mathbf{y}_i, \mathbf{p}_i). \qquad (3)$$

Here $\mathbf{y}_i$ is a one-hot vector representing the ground-truth label, and $\mathrm{CE}(\cdot, \cdot)$ is the cross-entropy loss. In the implementation of $\mathcal{L}_{\mathrm{VTM}}$, we apply online contrastive hard negative mining (Li et al., 2021b). We refer readers to the ALBEF paper (Li et al., 2021b) for a comprehensive introduction to momentum distillation and online contrastive hard negative mining.

**Captioning Loss.** This objective activates the video-grounded text decoder to predict the precise tokenized caption $c$ in an autoregressive way:

$$\mathcal{L}_{\mathrm{CAP}} = -\sum_{i=1}^{M}\log P\left(c_i \mid c_{<i}, V\right), \qquad (4)$$

where $M$ is the text length. Combining Eq. (2)-(4), the overall objective can be formulated as:

$$\mathcal{L} = \mathcal{L}_{\mathrm{VTC}} + \mathcal{L}_{\mathrm{VTM}} + \mathcal{L}_{\mathrm{CAP}}. \qquad (5)$$

### A.3 Hyperparameters.

The model is pre-trained for 5 epochs with the Adam (Kingma and Ba, 2015) with a weight decay of 5e-2. The batch size is 384 and the momentum queue size is 57600. The pre-training is conducted on four nodes with 32 NVIDIA V100 GPUs (32 GB memory per GPU) in total and each epoch lasts around 6 hours. The learning rate is linearly warmed up from 1e-6 to 5e-6 in the first 5000 steps and then gradually cosine decayed to 5e-7 in the remaining steps. Temporally consistent random spatial augmentation (Qian et al., 2021) is applied and mixed precision is used for efficient training.

## B Fine-tuning Details

The downstream fine-tuning is conducted on 8 NVIDIA V100 GPUs. The learning rate is 1e-5 with a warmup ratio of 0.1. The batch size is 16 and the momentum queue size is 32. We fine-tune our model for 10 epochs with the Adam optimizer and a weight decay of 0.05. For paragraph-to-video retrieval, we use $\mathcal{L}_{\mathrm{VTC}}$ and $\mathcal{L}_{\mathrm{VTM}}$ as training objectives. For evaluating paragraph-to-video retrieval models, we select the top 128 candidates based on the video-text feature similarity and then rerank the selected candidates by their pairwise VTM scores. For video-QA, we use the cross-entropy loss for maximizing the generation probability of the correct answer and rank the candidates by their generation probabilities for evaluation.

## C  Downstream Datasets

We finetune and evaluate TESTA on two downstream tasks of paragraph-to-video retrieval and long-form VideoQA. The details of these datasets are shown in Table 7.

For paragraph-to-video retrieval, we use 4 datasets of DiDeMo (Hendricks et al., 2017), QuerYD (Oncescu et al., 2020), ActivityNet Captions (Krishna et al., 2017), and Condensed Movie (Bain et al., 2020). We evaluate text-to-video retrieval, where the text acts as the query, in terms of R@$k$, which means the recall (%) of the target video through $K$ retrieval efforts.

For long-form VideoQA, we use ActivityNet-QA (Yu et al., 2019). The metric is accuracy (%).

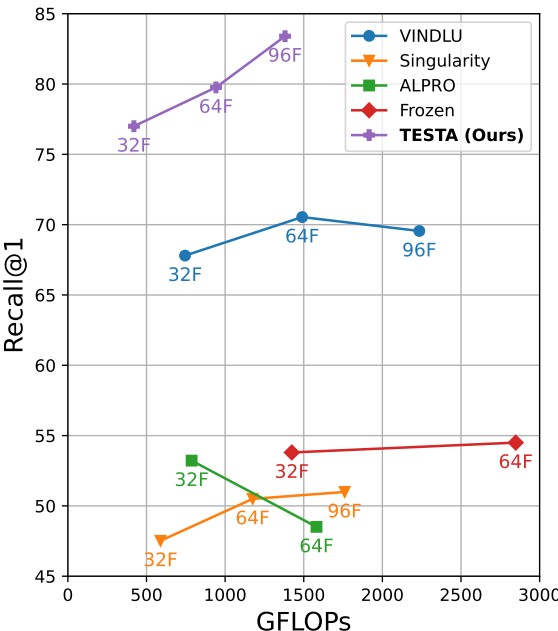

Figure 5: Comparison of GFLOPs and Recall@1 on the QuerYD dataset. $n$F denotes using $n$ input frames for fine-tuning and evaluation. The curve of our TESTA is located in the upper left corner, indicating that our model achieves a better performance-cost tradeoff compared to other pre-trained models.

## D  Recall-GFLOPs Tradeoff of Various Pre-trained Models

In Figure 7, we analyze the tradeoff between recall and GFLOPs for various pre-trained models. The curve of our TESTA is located in the upper left corner, indicating that our model achieves a superior Recall-GFLOPs tradeoff compared to other pre-trained models.

Furthermore, Figure 7 presents the model performance with different input frames. Surprisingly, increasing the number of input frames from 32 to 96 has minimal impact on the performance of Singularity (Lei et al., 2022) and Frozen (Bain et al., 2021), and even slightly reduced the recall of ALPRO (Li et al., 2021a) and VINDLU (Cheng et al., 2022). In contrast, our TESTA exhibits linear improvement in performance with the number of input frames, demonstrating superior scalability.

## E  Ablation on the Number of Reduced Tokens

In our TESTA (§ 3.2), $R_T$ and $R_S$ specify the number of tokens to be reduced for the temporal and spatial aggregation module, separately. To investigate the influence of these two hyper-parameters, we vary the number of $R_T$ and $R_S$, then report the average GFLOPs (blue bars) and recall (red star) on the QuerYD dataset. Figure 6 illustrates the results. On one hand, GFLOPs decrease linearly as $R^6$ increases, indicating that increasing the reduced token number can improve the efficiency of video encoding. On the other hand, merging too many tokens with large $R$ (e.g., $R_T = 10$) will lose semantic information in the final encoded video representation, thus leading to a declined average recall.

We evaluate more cases with various $R_T$ and $R_S$ configurations, and plot the GFLOPs-Recall tradeoff in Figure 7. Based on these results and analysis, we determined the default configuration for our TESTA, i.e., $R_T = 4$ & $R_S = 8$ and for 96-frame inputs, and $R_T = 1$ & $R_S = 12$ for 32-frame inputs. This configuration helps our model achieve approximately a $50\%$ reduction in computation cost without significant performance decline.

## F  Token Similarity Analysis

We probe the properties of the aggregated tokens by analyzing their similarity. In Figure 8, we count the average similarity between tokens from different blocks, different dimensions (frame tokens or patch tokens), and different aggregation results (aggregated or disaggregated).

**For patch tokens** (in orange), the overall similarity between them is large (higher than $0.5$), indicating considerable spatial redundancy. Meanwhile, the aggregated patch tokens (in dark orange) have a very high similarity of $0.96$, which ensures the semantic purity of the aggregated patch tokens.

---

[6]Here we use $R$ to refer to $R_T$ or $R_S$ for brevity.

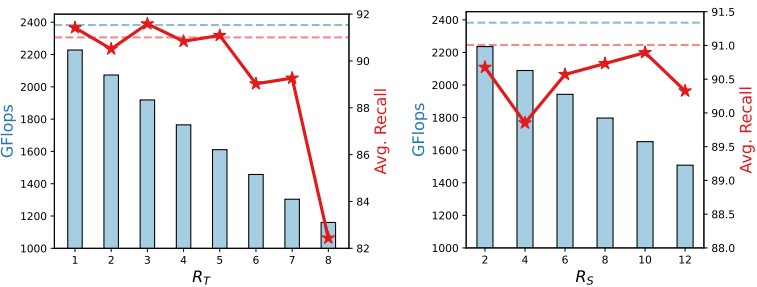

Figure 6: Ablation on reduced the token number, $R_T$ (temporal aggregation), and $R_S$ (spatial aggregation). The average recall is represented by red stars, while GFLOPs are depicted by blue bars. The dotted lines denote the results without any aggregation ($R_T = 0$ and $R_S = 0$). All results are evaluated on QuerYD with 96 frames.

Figure 7: GFLOPs-Recall tradeoff on QuerYD. We record the performance (dots) of TESTA with various $R_T$-$R_S$ configurations, and plot the trends (curve) by fitting the dots.

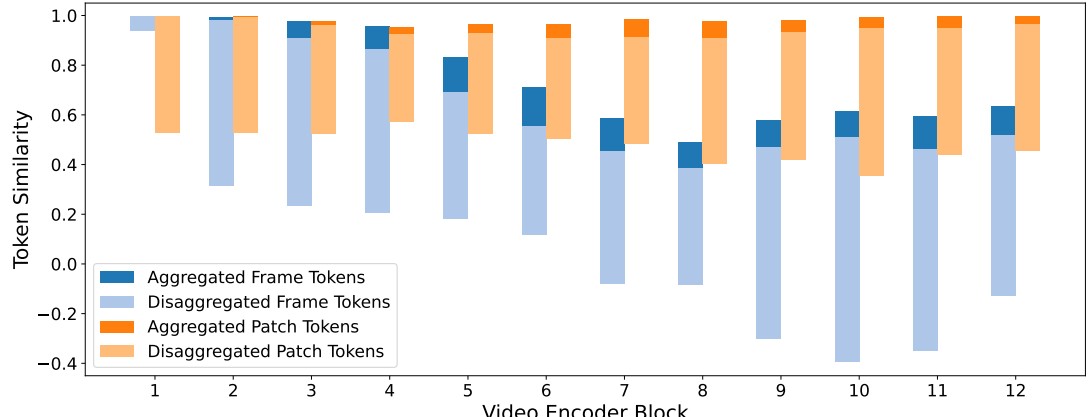

Figure 8: Cosine similarity between tokens from Set $\mathbb{A}$ and Set $\mathbb{B}$ in various video encoder blocks. The blue color indicates frame tokens while the orange color indicates patch tokens. For those tokens finally being aggregated, we plot their similarity in a dark color.

**While for frame tokens** (in blue), their similarity decreases as the number of blocks increases, which may yield aggregated frames with mixed and diverse semantics. Nevertheless, recall that our frame token is a pseudo token (§ 3.2.1) obtained by averaging patch features, which does not elaborately model frame semantics. Therefore, compared to patch tokens, the representation of frame token and their similarity measure needs improvement, which we regard as future work.

## G  More Visualization of Aggregation

In this section, we provide more qualitative results of our TESTA for video-language understanding. Figure 9 shows another 4 case on the DiDeMo dataset. TESTA effectively aggregates tokens with highly-similar semantics, demonstrating its strong interpretability.

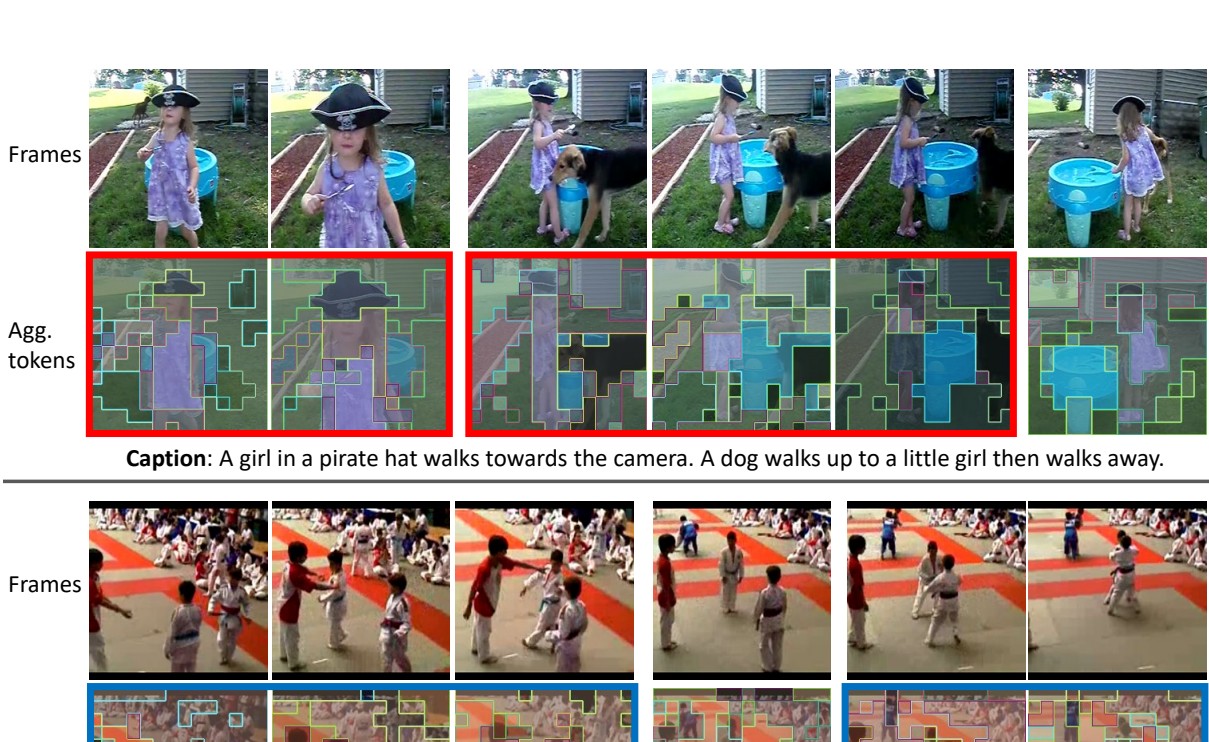

**Caption**: A girl in a pirate hat walks towards the camera. A dog walks up to a little girl then walks away.

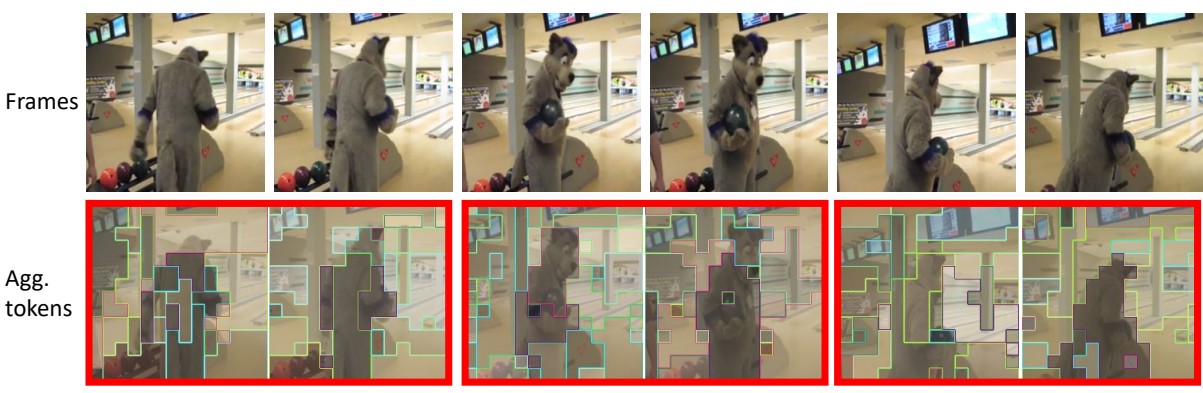

**Caption**: Two boys in a martial arts competition square off and bow to each other to start the match.

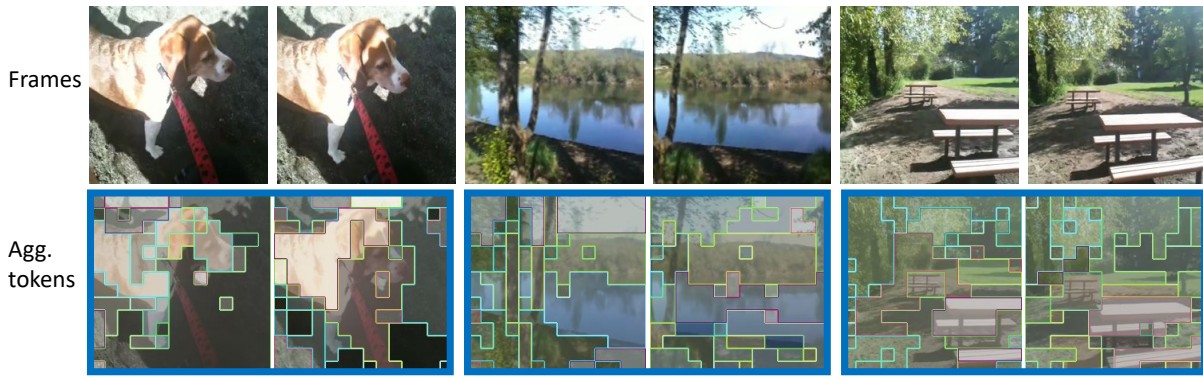

**Caption**: Man in a dog costume throwing a bowling ball.

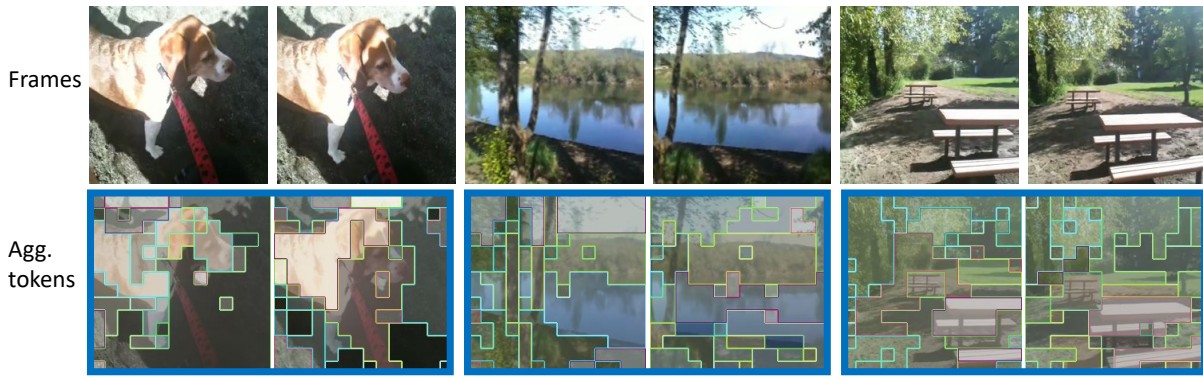

**Caption**: A dog sits at the beach. Camera pans from dog to water. There is a picnic table in the scene.

Figure 9: More visualizations of our aggregation on DiDeMo. Frames that are enclosed within the same rectangle, as well as patches that share the same inner and border color, are merged together.