# OpenReview forum: "TESTA: Temporal-Spatial Token Aggregation for Long-form Video-Language Understanding"
_EMNLP/2023/Conference — EMNLP 2023 Findings_

### Official Review · Reviewer_YtSY · 2023-07-26

**Soundness:** 2

**Excitement:**

2: Mediocre: This paper makes marginal contributions (vs non-contemporaneous work), so I would rather not see it in the conference.

**Paper Topic And Main Contributions:**

This paper proposes to aggregate visual tokens in text-video retrieval. They extend the ToMe[1] method to text-video retrieval.

[1] https://arxiv.org/pdf/2210.09461.pdf

**Questions For The Authors:**

- A. This paper is a direct extension of ToMe[1]. However, the author does not show enough respect for this previous work.
    - A1. Line 164 "previous work explored merging similar patches in the image". Actually, ToMe[1] also conducts video experiments in its Section 5 VIDEO EXPERIMENTS. A clear statement about the difference between the proposed method and ToMe is needed. Plus, a direct comparison in performance is also necessary.
   - A2. About why matching and not clustering, ToMe has a clear explanation in its Section 3 TOKEN MERGING. In 3.2.2 Aggregation Strategy of the paper, the author has a similar explanation without clearly pointing out that they are following ToMe.
   - A3. For the Geometry-based Aggregation, I cannot see any difference with the Aggregation strategy of ToMe. As for Importance-based Aggregation, it is much worse than Geometry-based Aggregation.
   - A4. Actually, it is not something new for aggregating tokens in text-video retrieval, for example, CenterCLIP[2].

- B. unfair comparison with previous work.
    - B1. The authors adopt the architecture and pre-trained weights of BLIP. It is necessary to provide the baseline in every table like Table 1.
    - B2. Please provide the inference time. GFLOPs cannot represent the speed.
    - B3. I am confused about the #PT Data. For example, Singularity[4] uses CLIP weights and its #PT Data is 5M, while Clip4Clip also uses CLIP weights and its #PT Data is 400M. The authors use BLIP weights, it pre-trained on more than 120M text-image pairs, but its #PT Data is 5M in Table 2. This will mislead readers. Plus, please provide the data type when showing #PT Data numbers, i.e., image or video.

- C. Please provide results on MSRVTT, both retrieval and video QA. Because these results are also reported in the original BLIP paper. The authors can also provide results with CLIP. It would be easier to judge the method if the CLIP results are available.

- D. Does the temporal aggregation will merge two discontinuous frames, for example, frame 1 and frame 31. Or, if frame 1 and frame 31 have a similar appearance, will they be merged? Will this broken the temporal relationship for some fine-grained video dataset, for instance, something-something dataset (see the github link of [4]).



[1] https://arxiv.org/pdf/2210.09461.pdf

[2] https://arxiv.org/pdf/2205.00823.pdf

[3] BLIP: Bootstrapping Language-Image Pre-training for Unified Vision-Language Understanding and Generation.

[4] Jie Lei, Tamara L. Berg, and Mohit Bansal. 2022. Revealing single frame bias for video-and-language learning. ArXivbs/2206.03428. https://github.com/jayleicn/singularity

**Reasons To Accept:**

This paper further validates that token aggregation also works for text-video retrieval.

**Reasons To Reject:**

1. Lack of technique contributions.
2. Lack of thorough analysis of the proposed method.



**Reproducibility:**

4: Could mostly reproduce the results, but there may be some variation because of sample variance or minor variations in their interpretation of the protocol or method.

**Reviewer Confidence:**

4: Quite sure. I tried to check the important points carefully. It's unlikely, though conceivable, that I missed something that should affect my ratings.

---

> ### Author Rebuttal · Authors · 2023-08-29
>
> Thanks for your insightful comments. We humbly think that some concerns are caused by misunderstandings, which we will explain in detail below. We hope that our response can clarify the misunderstandings so that you can consider our work more favorably.
>
> >**Q-A1: A clear statement about the difference between the proposed method and ToMe is needed. Plus, a direct comparison in performance is also necessary.**
>
> A-A1: Our paper is NOT a trivial extension of ToMe; rather, we differentiate ourselves from ToMe in two significant ways:
>
> **(1) Video Token Definition:** ToMe uses **joint space-time tokens** (2x16x16 cubes), while our TESTA defines **frame tokens** (representing entire frames) and **patch tokens** (16x16 2D patches) for decoupled aggregation. This tailored token design is more efficient for modeling long-form videos.
>
> **(2) Aggregation Method:** ToMe performs **global aggregation** over all tokens, resulting in complexity of $\mathcal{O}((\frac{T}{2} H W)^2)$ (T: frame number, H: patch number in height, W: patch number in width). This becomes impractical for long-form video and causes out-of-memory issues beyond 16 frames.  In contrast, TESTA uses **divided aggregation** in time and space, reducing complexity to $\mathcal{O}(T^2+(HW)^2)$. This allows efficient encoding of much longer videos (more than 128 frames under the same computation quota). The divided scheme also better captures spatial and temporal semantics, resulting in improved performance on long-form video understanding tasks.
>
> To directly compare performance, we initialized both ToMe and TESTA from the BLIP checkpoint and finetuned on QuerYD (16 frames). We set the hyperparameter R (number of reduced tokens) to ensure matched throughput. The results are as follows:
> | model |  |  QuerYD | | Throughput |
> |:----------|:---|:---|:---|:---:|
> | | R@1 | R@5 | R@10 | video/s |
> | ToMe | 59.9 | 82.2 | 88.6 | 4.56 |
> | TESTA | 64.1 (+4.2) |  85.9 (+3.7) | 90.1 (+1.5) | 4.64 |
> | ToMe (w/o agg.) | 66.1 | 86.4 | 90.4 | 1.56 |
> | TESTA (w/o agg.) | 75.0 (+8.9) | 91.1 (+4.7) | 93.8 (+3.4) | 1.52 |
>
> The results demonstrate TESTA's effectiveness for long-form video understanding, thanks to our specific design for divided spatial-temporal modeling. We are glad to incorporate this comparison in our revision.
>
> >**Q-A2: About why matching and not clustering**
>
> A-A2:  We believe it is evident that clustering methods, such as k-means, are not well-suited for parallelism due to their reliance on iterative algorithms. We will cite ToMe's findings to enhance the strength of our argument.
>
> >**Q-A3: For the Geometry-based Aggregation, I cannot see any difference with the aggregation strategy of ToMe**
>
> A-A3: As we stated in Section 3.2.2, our geometry-based aggregation is inspired by ToMe, nevertheless,
>
> **(1)** Our method incorporates decoupling for patch and frame tokens, which is tailored for long-form videos and brings much more efficiency (see A-A1).
>
> **(2)** Our experiments of geometry- and importance-based aggregation provides novel insights into the success factors of geometry-based approaches. We reveal adjacent tokens have higher similarity, indicating they should be separated into different sets during aggregation. This complements ToMe's principles by further elucidating why considering token position benefits aggregation performance.
>
> >**Q-A4: It is not something new for aggregating tokens in text-video retrieval, for example, CenterCLIP**
>
> A-A4: CenterCLIP aggregates video tokens based on clustering methods, which is much less efficient than our matching-based strategy. Furthermore, our TESTA outperforms CenterCLIP by a significant margin (e.g., 53.7 R@1 compared to 46.2) in ActivityNet retrieval. We will include a comparison with CenterCLIP in our revision.
>
> >**Q-B1: It is necessary to provide the baseline in every table like Table 1.**
>
> A-B1: Thanks for your suggestions. We provide the baseline results for Table 2 as follow:
> |  model  | #PT Data | #Frame | GFLOPs | | DiDeMo |  |  | ActivityNet|  |
> |:----------|:---:|:---:|:---:|:---|:---|:---|:---|:---|:---|
> |  |  | | | R@1 | R@5 | R@10 | R@1 | R@5 | R@10 |
> | TESTA | 5M | 32 | **420** | 56.6 | 81.4 | 87.8 | 51.7 | 79.1 | 87.6 |
> | TESTA, w/o agg. | 5M | 32 | 786 | 57.1 (+0.5) | 82.1 (+0.7) | 87.9 (+0.1) | 52.6 (+0.9) | 79.5 (+0.4) | 88.1 (+0.5) |
> | TESTA | 5M | 96 | **1381** | 59.2 |  83.5 | 89.8 | 53.7 | 79.9 | 88.9 |
> | TESTA, w/o agg. | 5M | 96 | 2383 |  59.5 (+0.3) | 83.7 (+0.2) | 90.0 (+0.2) | 53.9 (+0.2) | 80.0 (+0.1) | 88.9 (+0.0) |
>
> The results and conclusions align with that in Table 1: Although preserving full visual tokens (TESTA, w/o agg.) yields higher recall, it requires 1.8 times more GLFOPs compared to TESTA. As the number of input frames increases from 32 to 96, the GFLOPs of TESTA w/o agg. exceed 2300, but the performance gain diminishes (only +0.2 R@1 on ActivityNet). This indicates the superiority of our method in aggregating redundant tokens in long sequence inputs.
>
> >**Q-B2: Please provide the inference time. GFLOPs cannot represent the speed.**
>
> A-B2: We evaluate the throughput (video/s) of representative models based on 4xA40 GPUs (QuerYD with 64 frames). The results are as follow:
> | Model | R@1 | GFLOPs | Throughput (video/s) |
> |:---|:---:|:---:|:---:|
> | ALPRO | 48.5 | 1581 | 0.22 |
> | Singularity | 50.5 | 1175 | 0.76 |
> | VINDLU | 70.5 | 1490 | 0.73 |
> | TESTA | **79.8** | **942** | **1.21** |
>
> Our model achieves the best efficiency under both GFLOPs and throughput measures. We note that GFLOPs are commonly used as an efficiency measure in previous works like TimeSFormer, VINDLU, MViT, etc., as they are less affected by code implementation. We align with these works.
>
> >**Q-B3: I am confused about the #PT Data.**
>
> A-B3: **The #PT Data statistics in our paper are provided in the original papers and are consistent with previous work, such as VINDLU, HiTeA, etc**. In the case of CLIP4clip, it uses CLIP's pre-trained checkpoint and directly fine-tunes it on the downstream task, so the #PT Data is considered as CLIP's #PT Data (400M). As for Singularity, although it is also initialized from CLIP's checkpoint, it undergoes pre-training on 5M data, hence the #PT Data is considered as 5M.
>
> Similarly, our model is initialized from BLIP's checkpoint but undergoes pre-training on 5M data, resulting in our #PT Data being 5M. It's important to note that such statistics are normal since BLIP also initializes its model with ViT pre-trained on ImageNet and BERT, but the PT Data of ImageNet and BERT is not taken into account.
>
>
> >**Q-C: Please provide a direct comparison to BLIP and CLIP on MSRVTT dataset.**
>
> A-C: We provide the **zero-shot** performance of pre-trained CLIP4clip, BLIP and TESTA (TESTA w/o agg.) on MSRVTT (short-form videos) and QuerYD (long-form videos) datasets (32 frames). Additionally, we apply our token aggregation to pretrained TESTA in an off-the-shelf manner (TESTA w/ agg.).
> |  model  |  | MSRVTT |  |  | QuerYD |  | Throughput |
> |:----------|:---|:---|:---|:---|:---|:---|:---|
> | | R@1 | R@5 | R@10 | R@1 | R@5 | R@10 | video/s |
> | CLIP4clip | 33.8 | 55.7 | 64.8 | 50.0 | 74.5 | 83.3 | 2.49 |
> | | | | | | | |
> | BLIP    | 42.4 | 64.0 | 72.5 | 50.7 | 67.6 | 73.5 | 1.24 |
> | TESTA (w/o agg.) | 42.8 (+0.4) | 64.7 (+0.7) | 75.1 (+2.6) | 64.4 (+13.7) | 82.9 (+15.3) | 86.9 (+13.4) | 1.08 (0.9x) |
> | TESTA (w/ agg.) | 42.4 (+0.0) | 65.1 (+1.1) | 73.5 (+1.0) | 62.1 (+11.4) | 82.2 (+8.7) | 85.4 (+11.9) | 1.68 (1.4x) |
>
> The results show that **our pre-trained TESTA outperforms BLIP (as well as CLIP4clip), achieving +1.2 and +14.1 average improvements** on MSRVTT and QuerYD, respectively. Particularly, the gains on long-form video datasets (QuerYD) is quite significant due to our temporal modeling and pre-training on video data.
>
> **After applying our token aggregation, TESTA maintains a considerable performance advantage over BLIP (+0.7 and +7.0 Recall-mean),  and also increases throughput by 1.4x.** These results indicate that it is not trivial to directly apply BLIP, an image-text model, to long-form video understanding tasks. By leveraging our pre-training and token aggregation methods, we achieve both improved effectiveness and efficiency compared to BLIP.
>
> >**Q-D: Does temporal aggregation merge two discontinuous frames? Will this break the temporal relationship for some fine-grained video dataset?**
>
> A-D:  Yes, temporal aggregation can merge two discontinuous frames. However, this does not break the temporal relationship due to the following reasons: **(1)** Frame tokens contain time information through time embeddings, so temporal distance is considered along with visual similarity during frame aggregation. Thus, distant frames like frame 1 and frame 31 are not encouraged to be merged. **(2)** Even if two discontinuous frames are aggregated, their temporal information is partially preserved through mean pooling. **(3)** It's important to note that our experiments focused on coarse-grained and long-form datasets, where redundant visual contexts often appear across discontinuous frames (e.g., repeated appearances of hosts). These redundant contexts can be merged without negatively impacting performance.
>
> **We hope our response can address your concerns and we will revise our paper based on your suggestions. We would be grateful if you could reassess the paper in light of these clarifications and consider adjusting the scores accordingly.**

---

### Official Review · Reviewer_jb6J · 2023-07-29

**Soundness:** 3

**Excitement:**

3: Ambivalent: It has merits (e.g., it reports state-of-the-art results, the idea is nice), but there are key weaknesses (e.g., it describes incremental work), and it can significantly benefit from another round of revision. However, I won't object to accepting it if my co-reviewers champion it.

**Paper Topic And Main Contributions:**

This paper investigates the problem of video processing in video-to-language tasks. It argues that past video processing methods for maintaining computational efficiency in video-language learning have the following limitations: 1. Sparse sampling may miss the semantic information in the video. 2. Offline encoding does not transfer the model well to downstream tasks. To solve these problems, this paper proposes an end-to-end video language learning method considering sufficient input video frames.

**Questions For The Authors:**

1. Is there any experimental analysis to prove that offline encoding affects the adaptation to downstream tasks?
2. Are there any experiments to verify the utility of this proposed method on other sota models?

**Reasons To Accept:**

1. This paper proposes a method for solving problems of important practical significance in current video language tasks;
2. The method presented in this paper works well. The model is able to achieve comparable performance with its counterpart (without adding token aggregation modules), with a computational burden reduction of close to 50%;


**Reasons To Reject:**

1. Although this paper applies Token Merging to the field of video-language to fuse features from temporal and spatial dimensions, its novelty is still limited;
2. The results of the proposed method on VideoQA do not seem to be very impressive. For example, HiTeA, with much fewer GFLOPs, reported better performance on ActivityNet-QA than TESTA.
3. I think this work would be more significant if there were more experiments to prove that the proposed token aggregation is a universal module.
4. Is there any experimental analysis to prove that offline encoding affects the adaptation to downstream tasks? As far as I know, there are many methods based on using CLIP image encoder to extract frame features, and CLIP itself has strong generalization.

**Reproducibility:**

4: Could mostly reproduce the results, but there may be some variation because of sample variance or minor variations in their interpretation of the protocol or method.

**Reviewer Confidence:**

4: Quite sure. I tried to check the important points carefully. It's unlikely, though conceivable, that I missed something that should affect my ratings.

---

> ### Author Rebuttal · Authors · 2023-08-29
>
> Thank you for your thoughtful feedback. We humbly think that some concerns are caused by misunderstandings, which we will explain in detail below. We hope that our response can clarify the misunderstandings so that you can consider our work more favorably.
>
> >**Q1: The novelty is limited compared to Token Merging**
>
> A1: Compared to Token Merging (ToMe), our TESTA model offers two key innovations in:
>
> **(1) Video Token Definition.** ToMe uses **joint space-time tokens** (2x16x16 cubes), while our TESTA defines **frame tokens** (representing entire frames) and **patch tokens** (16x16 2D patches) for decoupled aggregation. This tailored token design is more efficient for modeling long-form videos.
>
> **(2) Aggregation Method:** ToMe performs **global aggregation** over all tokens, resulting in complexity of $\mathcal{O}((\frac{T}{2} HW)^2)$ (T: frame number, H: patch number in height, W: patch number in width). This becomes impractical for long-form video and causes out-of-memory issues beyond 16 frames.  In contrast, TESTA uses **divided aggregation** in time and space, reducing complexity to $\mathcal{O}(T^2+(HW)^2)$. This allows efficient encoding of much longer videos (more than 128 frames under the same computation quota). The divided scheme also better captures spatial and temporal semantics, resulting in improved performance on long-form video understanding tasks.
>
> To directly compare performance, we initialized both ToMe and TESTA from the BLIP checkpoint and finetuned on QuerYD (16 frames). We set the hyperparameter R (number of reduced tokens) to ensure matched throughput. The results are as follows:
> | model |  |  QuerYD | | Throughput |
> |:----------|:---|:---|:---|:---:|
> | | R@1 | R@5 | R@10 | video/s |
> | ToMe | 59.9 | 82.2 | 88.6 | 4.56 |
> | TESTA | 64.1 (+4.2) |  85.9 (+3.7) | 90.1 (+1.5) | 4.64 |
> | ToMe (w/o agg.) | 66.1 | 86.4 | 90.4 | 1.56 |
> | TESTA (w/o agg.) | 75.0 (+8.9) | 91.1 (+4.7) | 93.8 (+3.4) | 1.52 |
>
> The results clearly demonstrate TESTA's effectiveness for long-form video understanding, thanks to our specific design for divided spatial-temporal modeling. We would gladly incorporate this comparison in our revision.
>
>
> >**Q2: The results of the proposed method on VideoQA do not seem to be very impressive compared to HiTeA**
>
> A2: In terms of efficiency, HiTeA utilizes MViTv2 [1] as a video encoder, which employs convolution-based pooling modules for feature downsampling. However, as we stated in Introduction, **this has two drawbacks compared to our approach: (1)** It introduces additional parameters (convolution kernels), requiring extra training. **(2)** Our token aggregation approach offers better interpretability as it allows for easier tracking and recovery of trajectories.
>
> Regarding performance on VideoQA, HiTeA incorporates two additional pre-training objectives (Moment & Temporal Relation Exploration) that benefit QA on specific moments. It achieves comparable performance to our TESTA (45.1 vs 45.0). We believe these pre-training techniques are compatible with our method and can be applied to further enhance our model, which we leave for future work. It is worth noting that our TESTA significantly outperform HiTeA on video retrieval tasks (+4.8 on DiDeMo and +6.6 ANet).
>
> [1] MViTv2: Improved Multiscale Vision Transformers for Classification and Detection. Li et al. CVPR 2022.
>
> >**Q3: I think this work would be more significant if there were more experiments to prove that the proposed token aggregation is a universal module.**
>
> A3: Thanks for your suggestions. We apply our token aggregation technique to ALPRO [1] and perform finetuning on QuerYD (32 frames). The results are as follow:
> | model | R@1 | R@5 | R@10 | GFLOPs  |
> |:----------|:---|:---|:---|:---:|
> | ALPRO | 50.7 | 76.5 | 84.9 | 788 |
> | ALPRO w/ our agg. | 49.5 | 74.8 | 84.4 | 421 |
>
> The results indicate that by incorporating our token aggregation, ALPRO achieves comparable performance while reducing the computational cost (GFLOPs) to 53% of the original amount. This demonstrates the effectiveness and universality of our approach. In future work, we plan to validate our approach on more models to further support its universality.
>
> [1] Align and Prompt: Video-and-Language Pre-training with Entity Prompts. Li et al. CVPR 2022.
>
> >**Q4: Is there any experimental analysis to prove that offline encoding affects the adaptation to downstream tasks?**
>
> A4: Yes. Our intention is to convey that offline encoding relies on post-aggregation mechanisms, which prohibits full temporal fusion at the beginning. VINDLU [1] conducted experiments in its Section 3 Step 1, revealing that **post-aggregation methods** (mean pooling / late temporal attention) **perform significantly worse (-5.7 acc. on average) compared to full temporal-spatial modeling methods** (Temporal Convolution / Temporal Attention at each layer). Consequently, previous works that directly adapt CLIP's image encoder (e.g., CLIP4clip, XCLIP) often yield inferior results compared to approaches utilizing video encoders with full spatial-temporal modeling (e.g., HiTeA, VINDLU, and our TESTA).
>
> [1] VINDLU: A Recipe for Effective Video-and-Language Pretraining. Cheng et al .CVPR 2023
>
> >**Q5: Are there any experiments to verify the utility of this proposed method on other sota models?**
>
> A5: Yes, we have applied our method to ALPRO, and the results confirm its utility, please refer to A3.
>
> **We hope our response can address your concerns and we will revise our paper based on your suggestions. We would be grateful if you could reassess the paper in light of these clarifications and consider adjusting the scores accordingly.**

---

### Official Review · Reviewer_E2NS · 2023-08-05

**Typos Grammar Style And Presentation Improvements:** The presentation is good.
**Soundness:** 4

**Excitement:**

4: Strong: This paper deepens the understanding of some phenomenon or lowers the barriers to an existing research direction.

**Missing References:**

The paper discusses several benchmarks on long-video e.g., activitynet, and condensed movies; but they are actually a few minutes long, while the real-world long videos may be several hours e.g., untrimmed egocentric videos (Ego4D), whole movie (MAD). which are more challenging, and should be mentioned in the paper.

**Paper Topic And Main Contributions:**

This paper proposes an efficient end-to-end video-language pretraining framework for long-form videos. Especially, the paper proposes a novel strategy to adaptively aggregate relevant frames and patches, which can significantly reduce the computational cost of visual tokens. Besides, they propose a VLP model with a divided space-time token aggregation.
The paper validates their components and reaches satisfactory performance on paragraph-to-video retrieval and long-form VideoQA benchmarks, and provides strong visualization to demonstrate their designs.

**Questions For The Authors:**

1. Please refer to *Reasons To Reject.
2. Will you release the code to community?

**Reasons To Accept:**

1. The long-form video-language pretraining is a valuable yet challenging problem, which is less explored compared to common short-term videos.
2. The paper proposes a novel aggregation strategy to overcome the heavy visual tokens computations. Notably, they use the bipartite matching method to merge the tokens based on their attention key features. They further design two variants to study the aggregation.
3. The benchmark results over long-video tasks, demonstrate their methods can achieve good efficiency while maintaining strong performance.
4. Clear ablation studies with convincing numbers and good visualization,  validate their designs.

**Reasons To Reject:**

1. In Fig.2, it appears to have three text encoder variants, mainly with shared layers but with slightly different ones. will it be possible to unify as one text encoder but just with different switch options? e.g., FIBER?
2. Despite the paper achieving strong results on the benchmark, it relies on BLIP initialization, which should be discussed and ablation in experiments, how percentage improvement achieve over the image-text pretraining?
3. Why k-means is not suited for parallelism but the bipartitle matching method is ok? how's the computation cost of bipartite operation?


**Reproducibility:**

3: Could reproduce the results with some difficulty. The settings of parameters are underspecified or subjectively determined; the training/evaluation data are not widely available.

**Reviewer Confidence:**

4: Quite sure. I tried to check the important points carefully. It's unlikely, though conceivable, that I missed something that should affect my ratings.

---

> ### Author Rebuttal · Authors · 2023-08-29
>
> Thanks for your positive assessment and constructive feedback.
>
> >**Q1: Will it be possible to unify as one text encoder but just with different switch options?**
>
> A1: That's an intriguing idea. Currently, our model comprises two text encoder variants: Text Encoder and Video-grounded Text Encoder. These encoders serve different purposes, such as lightweight pre-ranking and fine-grained re-ranking for retrieval inference, respectively (Appendix B). The separation of these encoders simplifies code implementation and aligns with established practices in prior works like ALBEF and BLIP. However, we acknowledge the potential benefits of a unified encoder with switch options and will explore this direction in future research.
>
> >**Q2: TESTA's pre-training relies on BLIP initialization. How much improvement does TESTA achieve over BLIP?**
>
> A2: We provide the **zero-shot** performance of pre-trained BLIP and pre-trained TESTA (TESTA w/o agg.) on MSRVTT (short-form videos) and QuerYD (long-form videos). Additionally, we apply our token aggregation to pretrained TESTA in an off-the-shelf manner (TESTA w/ agg.). The table below summarizes the findings:
> |  model  |  | MSRVTT |  |  | QuerYD |  | Throughput |
> |:----------|:---|:---|:---|:---|:---|:---|:---|
> | | R@1 | R@5 | R@10 | R@1 | R@5 | R@10 | video/s |
> | BLIP    | 42.4 | 64.0 | 72.5 | 50.7 | 67.6 | 73.5 | 1.24 |
> | TESTA (w/o agg.) | 42.8 (+0.4) | 64.7 (+0.7) | 75.1 (+2.6) | 64.4 (+13.7) | 82.9 (+15.3) | 86.9 (+13.4) | 1.08 (0.9x) |
> | TESTA (w/ agg.) | 42.4 (+0.0) | 65.1 (+1.1) | 73.5 (+1.0) | 62.1 (+11.4) | 82.2 (+8.7) | 85.4 (+11.9) | 1.68 (1.4x) |
>
> The results show that **our pre-trained TESTA outperforms BLIP, achieving +1.2 and +14.1 average improvements** on MSRVTT and QuerYD, respectively. Particularly, the gains on long-form video datasets (QuerYD) is quite significant due to our temporal modeling and pre-training on video data.
>
> **After applying our token aggregation, TESTA maintains a considerable performance superiority over BLIP (+0.7 and +7.0 Recall-mean), and also increases throughput by 1.4x**. These results indicate that it is not trivial to directly apply BLIP, an image-text model, to long-form video understanding tasks. By leveraging our pre-training and token aggregation methods, we achieve both improved effectiveness and efficiency compared to BLIP.
>
> >**Q3: Why k-means is not suited for parallelism but the bipartite matching method is ok? How's the computation cost of bipartite operation?**
>
> A3: Our goal is to reduce N tokens to N-R tokens. **K-means**, being an iteration-based algorithm, requires K iterations to find N-R clusters. In each iteration, it computes the similarity between N tokens and N-R clusters. As a result, its computational complexity is $\mathcal{O}(KN(N-R))$.
>
> On the other hand, the **bipartite matching** method assigns R tokens to the remaining N-R tokens without iterations. It only needs to calculate the similarity between R tokens and N-R clusters. Consequently, its computational complexity is $\mathcal{O}(R(N-R))$, making it significantly more efficient than the k-means method.
>
> >**Q4: Will you release the code to the community?**
>
> A4: Absolutely, we are committed to releasing both the code and the pre-trained & fine-tuned checkpoints to the community.
>
> >**Q5: The real-world long videos like Ego4D should be mentioned in the paper.**
>
> A5: Thanks for your suggestions. We will cite these work in our revision.
>
> **We hope our response can address your concerns and we will revise our paper based on your suggestions.**

---

### Meta-Review · Area_Chair_YaW4 · 2023-09-20

**Recommendation:** 3

**Metareview:**

The core idea of this work is to aggregate similar frames each video and similar patches within each frame, with bipartite matching, to reduce the computation cost of processing long-form videos. Experiments on paragraph-to-video captioning and long-form videoQA, demonstrates the efficiency and performance gain of the approach. While the idea of token merging has been explored previously in ToMe, as mentioned by the reviewers, the adaption of it to long-form video processing requires specific designs such as decoupling frame-level and patch-level aggregation. Thus there is still merit of the approach to the community. One missing point of the work is to test the approach against ToMe on pure video tasks, since the approach works for videos alone as well. AC encourages the authors to provide experiments to make the work in a more complete form.

---

### Decision · Program_Chairs · 2023-10-07

**Decision:**

Accept-Findings

**Comment:**

The core idea of this work is to aggregate similar frames each video and similar patches within each frame, with bipartite matching, to reduce the computation cost of processing long-form videos. Experiments on paragraph-to-video captioning and long-form videoQA, demonstrates the efficiency and performance gain of the approach. While the idea of token merging has been explored previously in ToMe, as mentioned by the reviewers, the adaption of it to long-form video processing requires specific designs such as decoupling frame-level and patch-level aggregation. Thus there is still merit of the approach to the community. One missing point of the work is to test the approach against ToMe on pure video tasks, since the approach works for videos alone as well. AC encourages the authors to provide experiments to make the work in a more complete form.